# Benefits of Robotic Assisted vs. Traditional Laparoscopic Partial Nephrectomy: A Single Surgeon Comparative Study

**DOI:** 10.3390/jcm11236974

**Published:** 2022-11-26

**Authors:** Gerard Bray, Arya Bahadori, Derek Mao, Sachinka Ranasinghe, Christopher Tracey

**Affiliations:** 1Urology Department, Gold Coast University Hospital, Gold Coast, QLD 4215, Australia; 2Department of Medicine, Bond University, Gold Coast, QLD 4226, Australia

**Keywords:** uro-oncology, partial nephrectomy, renal cell carcinoma, robotics

## Abstract

Purpose: The current study aims to compare peri-operative and post-operative outcomes between robotic assisted vs. laparoscopic partial nephrectomy. Multiple reviews of the current literature have detailed the lack of single surgeon studies in this domain. To limit inter-operator bias, we utilise a single surgeon experienced in both approaches to reduce this bias seen in other multi-centre studies. Methods and Materials: We retrospectively compared patient demographics, tumour characteristics, peri-operative and post-operative outcomes of all partial nephrectomies undertaken by a single surgeon between 2014 and 2021 with experience in both laparoscopic and robotic surgery. The Da Vinci surgical system was utilized. Statistical analysis was carried out using GraphPad prism software version 7.03, San Diego, CA, USA. Results: Warm ischemia time was reduced by 2.6 min, length of stay reduced by 1.3 days and acute renal function deterioration was reduced by 55% with all these results being significant with robotic assisted partial nephrectomy compared to laparoscopic partial nephrectomy. Conclusion: This study highlights the benefits of robotic assisted in comparison to laparoscopic partial nephrectomy. Further large-scale prospective studies and cost-benefit analysis of robotic assisted partial nephrectomy would be valuable in confirming these findings and justifying the usage against their financial cost.

## 1. Introduction

Based on the current oncological and quality of life evidence, the European Association of Urology (EAU) currently recommends partial nephrectomy as the preferred management for localised cT1 renal tumours, irrespective of surgical approach. Emerging data also demonstrates partial nephrectomy to be safe in cT2 renal neoplasms as well [1]. The goals of partial nephrectomy are renal function preservation, negative margins and avoidance of peri-operative complications. Since these recommendations, open PN was generally the approach utilised due to operator comfort; however, with the advancements in laparoscopic technology and operator skills, the laparoscopic approach became the procedure of choice. Compared to open PN, the laparoscopic approach has comparable oncological outcomes with less morbidity and generally a faster recovery from surgery.

Both robotic assisted PN (RAPN) and traditional laparoscopic PN (LAPN) have steep operator learning curves [2]. Current systematic reviews of the literature have not shown clear differences in peri-operative and oncological outcomes between the two approaches [3]. There is some evidence that RAPN may reduce warm ischemia time, but this is not consistently seen [4]. Limitations from the recent systematic reviews indicate these comparative studies are often heterogenous, use small sample sizes (<50 patients) and often the results being user dependent [5]. Porpiglia et al. 2016, who carried out a large systematic review comparing the two approaches, highlighted the significant lack of single surgeon studies in the current literature [6]. 

There have only ever been two previous single surgeon studies carried out in this domain comparing RAPN with LAPN [4,7]. Both studies were carried out over ten years ago when robotic surgery was in its infancy and findings were inconsistent. Our study contributes a more recent single surgeon study comparing peri-operative outcomes of RAPN in a decade where robotics is more familiar and utilised. The current study aims to assess the differences in outcomes between the two approaches.

## 2. Materials and Methods

### 2.1. Study Design

We retrospectively reviewed all robotic assisted and traditional laparoscopic partial nephrectomy cases from a tertiary hospital. There was no exclusion based on patient factors or tumour factors. Analysis included all robotic and laparoscopic partial nephrectomies that took place between June 2017 and June 2021. The Da Vinci surgical system was utilised for this study. Patient demographics, peri-operative and post-operative outcomes were assessed. All cases were carried out by a urologist with over ten years’ experience in the field (10–13 cases each year). All data obtained for analysis was approved by the Human Research Ethics Committee (HREC), Canberra, Australia.

### 2.2. Outcomes

Patient demographics assessed included age, weight, gender, pre-operative renal function calculated by estimated glomerular filtration rate (GFR), mean tumour size/location (based on pre-operative computerised tomography [CT] imaging) and medical/surgical co-morbidities (hypertension, cardiovascular disease, and diabetes). The complexity of renal tumours was calculated using RENAL nephrometry scores, which take into account tumour radius, exophytic position, nearness to collecting system, anterior/posterior location and location to polar lines. Peri-operative outcomes included operation time, blood loss, haemoglobin drop and warm ischaemic time (hilar vessel clamp time). Post-operative outcomes assessed included length of stay, significant post-operative bleed defined as those needing embolization or haematoma on imaging and change in GFR both at day 1 and after 3 months.

### 2.3. Statistical Analysis

All numerical results from each outcome and group were combined for statistical analysis using GraphPad Prism (version 7.03), San Diego, USA. Comparisons in outcomes between both groups were performed using Mann–Whitney (unpaired, non-parametric) U tests. Chi squared test and Fisher’s exact test were used for bivariate analysis. A *p* value of <0.05 was considered statistically significant.

## 3. Results

A total cohort of 95 patients underwent LAPN (N = 49) or RAPN (N = 46) between the dates June 2017 and June 2021. All cases had a radiologically diagnosed renal mass, which was targeted for resection.

### 3.1. Patient Demographics

Patient demographics between both groups are summarized in Table 1. The robotic and laparoscopic cohorts had similar age, co-morbidities and pre-operative renal function. No statistically significant difference was found between either group in these domains. Prevalence of diabetes and hypertension tended to be higher in the laparoscopic group; however, cardiovascular disease tended to be more prevalent in the robotic group. The robotic cohort was found to have a higher percentage of males compared to the laparoscopic cohort (65% and 51% male, respectively). 

### 3.2. Tumour Characteristics

The radiological location and size of renal masses are summarised in Table 2. Post-operative histological sections were used for diagnosis of tumours and are also summarised in Table 2. There was no significant difference in the mean tumour size between groups (2.5 cm vs. 2.6 cm). There was no significant difference in nephrometry scores between either group. Scores ranged between low to moderate complexity with mean scores totaling 6.4 vs. 6.1 in RAPN and LAPN, respectively. Clear cell RCC was the most common tumour type in both groups. The robotic group had a larger proportion of benign lesions excised compared to the laparoscopic group (30% vs. 16%).

### 3.3. Peri-Operative Outcomes

The peri-operative outcomes measured are summarised in Table 3. Skin-to-skin mean operative time was significantly reduced in the RAPN group compared to the LAPN group. RAPN significantly reduced operative time by 15.2 min (141.8 vs. 157.0, *p* = 0.02). Mean blood loss was not always recorded in operative notes and, therefore, was likely under-estimated in both groups. Despite this, blood loss in the RAPN cohort was reduced compared to the LAPN group (15.5 vs. 29.4, *p* = 0.43). Warm ischemia time (clamp time) was significantly reduced by 2.5 min in the RAPN group compared with the LAPN group (13.9 vs. 16.4, *p* = 0.0046).

### 3.4. Post-Operative Outcomes

Post-operative outcomes for the RAPN and LAPN groups are summarised in Table 4. Length of stay (LOS) of patients was significantly reduced by 1.3 days in the RAPN group compared to LAPN groups (2.4 vs. 3.7, *p* = 0.0013). Bleeding complication rates were similar between both groups. Post-operative bleeding rates were diagnosed with CT imaging post-operatively showing a new haematoma or requirement for embolisation. Post-operative bleed requiring PRBC was lower in the RAPN compared to the LAPN groups (7% vs. 10%). Post-operative bleeding requiring embolisation was also similar between RAPN and LAPN groups (7% vs. 4%). Acute renal function deterioration was calculated from the difference in pre-operative glomerular filtration rate (GFR) and day 1 (D1) post-operative GFR. Chronic renal function deterioration was calculated from the difference in pre-operative GFR and the GFR taken 3 months post-operatively when GFR tends to plateau following surgery. Acute D1 renal function deterioration was significantly less in the RAPN group compared to the LAPN group (6.1 vs. 13.5, *p* = 0.029). There was no significant difference in chronic renal function deterioration between either group; however, the RAPN group tended to have a more preserved renal function long-term (9.1 vs. 10.7, *p* = 0.81). 

## 4. Discussion

The current standard of care recommends PN for T1 renal masses. PN has similar oncological outcomes to RN, while sparing the kidney and renal function. This becomes extremely useful in patients with single kidneys, poor renal function or renal masses that have not had prior biopsy and may represent benign or indolent tumour. The advent of laparoscopic technology has led to a push towards using this minimally invasive approach to carry out PN. LAPN is proven to be a safe alternative to open PN with comparable oncological outcomes and improved cosmetic and functional outcomes and reduced hospital stays [8]. The main limitations of LAPN include the increased risk of operative complications and the requirement of advanced laparoscopic skills. Another significant drawback is the need for clamping of hilar vessels and creating a warm ischemia in the renal tissues for the duration of renal tumour excision and renorrhaphy. Current studies in the field indicate a warm ischemia time over 30 min is associated with renal atrophy and irreversible ischaemic insult to the parenchyma [9]. 

Robotic surgery represents an evolving and expanding area in many surgical specialties including urology. In 2000, the Da Vinci surgery system became the first FDA approved robotic surgical device to be utilised for laparoscopic operations. Robotics allowed for high resolution, three-dimensional magnified vision, improved dexterity and less tissue contact to improve infection risk. Regarding PN, there is currently a need for ongoing studies to assess if these perceived benefits of robotics can translate into clinically significant improvements in intra-operative and post-operative outcomes compared to the traditional laparoscopic approach.

Comparisons between RAPN and LAPN have certainly been described in the current literature. Multiple systematic reviews in the field indicate RAPN to be a safe and effective alternative to LAPN. There is evidence that RAPN can reduce warm ischemia time compared to LAPN [3]. Additionally, the benefits of RAPN may be more pronounced in complex renal tumours. The advanced technologies able to be utilised by robotics, such as three-dimensional augmented reality and indocyanine green, are separating robotics from the traditional laparoscopic equipment [10,11]. Long et al. found RAPN was associated with significantly reduced conversion to radical nephrectomy and also reduced deterioration in GFR post-operatively compared to LAPN in complex tumours with high nephrometry scores (≥7) [12]. The current literature has been useful in displaying some of the benefits in RAPN; however, when comparing the two approaches, there remains inherent inter-operator bias in all the large-scale trials.

In the current study, we demonstrated that RAPN is comparable to LAPN with some end points showing significant superiority. We demonstrated that RAPN is associated with quicker operating time and reduced warm ischemia time. These intra-operative benefits are also illustrated in other studies. Benway et al. (2009) highlighted that the use of RAPN reduced warm ischemia time by 9.9 min (15.3 vs. 25.2 min), a much larger difference compared to our study [13]. This study also showed a significant reduction in blood loss, not reflected as noticeably as in ours. In regard to warm ischemia time, once below 30 min, the benefits to renal function of reduced clamp time are not clear. Our study demonstrated a reduced clamp time correlated with significantly improved preservation of renal function at day 1, which has not previously been described. This renal preservation did not seem to translate long-term as both groups reported similar renal function changes after 3 months. 

RAPN was also associated with significantly reduced length of stay in hospital compared to the LAPN cohort. Both approaches had acceptable bleeding complication rates and no significant difference was found in intra-operative blood loss or post-operative significant bleeding. Blood loss in the current study was likely under-estimated due to some incomplete documentation; however, similar findings have been seen in other reports [13]. These findings contrast multiple other earlier studies that did not find any benefit in post-operative outcomes between the two approaches [14]. We theorize that many of these earlier studies did not see clinically significant improvements in outcomes as robotic surgery was in its infancy during these years. As highlighted by multiple other more recent studies, we see the gap in outcomes between approaches widen as time and user experience grows. The benefits of RAPN highlighted by this paper may only be of modest proportions compared to some of the advantages that robotics may bring to renal sparing surgery in the future as experience grows amongst surgeons.

The leading limitation of the present study was the retrospective nature, which lends the paper to inherent bias and incomplete data collection, for example that seen when reporting operative blood loss volumes. Sample size is an inherent difficulty in single surgeon studies. While our sample was deemed adequate for statistical analysis, a larger sample could improve the quality of the paper. We also note that comparing operative times between different approaches, which consist of differing docking times, is not always the most accurate way to evaluate proficiency. Lastly, we also acknowledge that a single surgeon is a limitation due to smaller sample sizes and reduced reproducibility of the findings. Further large-scale prospective trials with inclusion of cost analyses would be useful in not only further elucidating the advantages of robotic surgery but justifying its usage against its financial cost.

## 5. Conclusions

In this single surgeon study, RAPN is a safe and effective approach, which in our study displayed multiple benefits compared to LAPN. Benefits of RAPN include quicker operative times, reduced clamp time, earlier hospital discharge and improved acute renal function. This represents the first single surgeon study to highlight these advantages in the modern era of robotics. Further large-scale prospective trials are required to further explore the role of robotics in renal preservation surgery.

## Figures and Tables

**Table 1 jcm-11-06974-t001:** Patient demographics, co-morbidities and renal function of those undergoing either laparoscopic or robotic partial nephrectomy.

	Robotic [N = 46] (%)	Laparoscopic [N = 49] (%)	*p*-Value
Age, mean	63.3	62.1	*p* = 0.42
<55	7 (15)	11 (22)	
55–64	16 (35)	16 (31)	
>64	23 (50)	22 (43)	
Gender			
Male	30 (65%)	25 (51%)	*p* = 0.21
Co-morbidities			
Hypertension	20 (43)	29 (59)	*p* = 0.15
Cardiovascular disease	8 (17)	4 (8)	*p* = 0.22
Diabetes	6 (13)	12 (24)	*p* = 0.19
Mean pre-op GFR (mL/min/1.73 m^2^)	80	78.8	*p* = 0.84

**Table 2 jcm-11-06974-t002:** Tumour characteristics: Tumour size and location based on pre-operative radiological imaging with either CT imaging or magnetic resonance imaging. Histological diagnosis from pathology specimens of lesion post-operatively.

	Robotic [N = 46] (%)	Laparoscopic [N = 49] (%)	*p* Value
Mean tumour size (cm)	2.5 (Max: 6.2)	2.6 (Max 5.6)	*p* = 0.78
Tumour location (%)			
Side	Right: 22 (48) Left: 24 (52)	Right: 26 (53) Left: 23 (47)	
Upper pole	20 (43)	16 (33)	
Inter polar	13 (28)	18 (37)	
Lower pole	13 (28)	15 (31)	
Mean renal nephrometry scores	6.4	6.1	*p* = 0.69
Tumour Type			
Clear cell RCC	17 (37)	24 (49)	
Papillary Type 1	7 (15)	8 (16)	
Papillary Type 2	4 (9)	4 (8)	
Chromophobe	4 (9)	3 (6)	
Other	0 (0)	2 (4)	
Benign	14 (30)	8 (16)	

**Table 3 jcm-11-06974-t003:** Peri-operative outcomes of robotic assisted and laparoscopic assisted partial nephrectomy groups.

	Robotic [N = 46]	Laparoscopic [N = 49]	*p*-Value
Mean operative time (min) [Hours]	141.8 [2:21]	157.0 [2:37]	*p* = 0.02 *
Mean blood loss (ml)	15.5	29.4	*p* = 0.43
Mean clamp time (min)	13.9	16.5	*p* = 0.0046 **

* *p* < 0.05, ** *p* < 0.01.

**Table 4 jcm-11-06974-t004:** Post-operative outcomes of robotic and laparoscopic partial nephrectomy cohorts.

	Robotic [N = 46] (%)	Laparoscopic [N = 49] (%)	*p*-Value
Mean length of stay (days)	2.4	3.7	*p* = 0.0013 **
Bleed requiring PRBC	3 (7)	5 (10)	*p* = 0.72
Bleed requiring embolization	3 (7)	2 (4)	*p* = 0.67
Acute (D1) GFR reduction (%)	6.1	13.5	*p* = 0.029 *
Long-term (3 months) GFR reduction	9.1	10.7	*p* = 0.81

* *p* < 0.05, ** *p* < 0.01.

## Data Availability

Data can be accessed on request by corresponding author.

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
