# Peer review of "Benefits of Robotic Assisted vs. Traditional Laparoscopic Partial Nephrectomy: A Single Surgeon Comparative Study"

_jcm, 2022, doi:10.3390/jcm11236974_

Round 1

Reviewer 1 Report

Dear Editor, thank you for giving me the opportunity to revise this paper. I read the manuscript with interest, since the topic is still debated. In general, the article is well written and the objectives are clear.

In order to consider this paper for publication, a major revision is needed. 

Here my remarks.

- INTRODUCTION: 

 “Based on the current oncological […] partial nephrectomy as the preferred management for localised cT1 renal tumours, irrespective of surgical approach.” This is absolutely true and also selected T2 renal neoplasms beneficiate from minimally invasive partial nephrectomy. About this point, I would cite the following article by a solid and recent single-surgeon experience: 

- Amparore D, Pecoraro A, Piramide F, Checcucci E, DE Cillis S, Volpi G, Piana A, Verri P, Granato S, Sica M, Manfredi M, Fiori C, Porpiglia F. Comparison between minimally-invasive partial and radical nephrectomy for the treatment of clinical T2 renal masses: results of a 10-year study in a tertiary care center. Minerva Urol Nephrol. 2021 Aug;73(4):509-517. doi: 10.23736/S2724-6051.21.04390-1. Epub 2021 Apr 22. PMID: 33887896.

- Ref. 1 and 6 are the same. Please delete one of them.

- The surgical complexity of the renal masses was not indicated. This is of primary importance when reporting surgical and post-operative results of partial nephrectomy in order to correctly interpret data.  Please stratify the tumor lesions according to their complexity.

- Mean skin-to-skin operative time was described lower in RAPN group. Robot docking is usually a time-consuming procedure and, in expert hands, laparoscopic procedures are generally faster if we consider the global operative time, especially for low-complexity renal masses.

- Bibliography contains mainly non up to date citations. I would suggest to implement bibliography with some newer and pertinent articles. For example, in the discussion, the possible advantages from robotics over laparoscopy may come especially for complex renal masses. In these cases, the use of new technologies may be game-changing in the improvement of robotic performances. For this reason, I would cite these renowned articles in the discussion: 

Porpiglia F, Checcucci E, Amparore D, Piramide F, Volpi G, Granato S, Verri P, Manfredi M, Bellin A, Piazzolla P, Autorino R, Morra I, Fiori C, Mottrie A. Three-dimensional Augmented Reality Robot-assisted Partial Nephrectomy in Case of Complex Tumours (PADUA ≥10): A New Intraoperative Tool Overcoming the Ultrasound Guidance. Eur Urol. 2020 Aug;78(2):229-238. doi: 10.1016/j.eururo.2019.11.024. Epub 2019 Dec 30. PMID: 31898992.

Amparore D, Checcucci E, Piazzolla P, Piramide F, De Cillis S, Piana A, Verri P, Manfredi M, Fiori C, Vezzetti E, Porpiglia F. Indocyanine Green Drives Computer Vision Based 3D Augmented Reality Robot Assisted Partial Nephrectomy: The Beginning of "Automatic" Overlapping Era. Urology. 2022 Jun;164:e312-e316. doi: 10.1016/j.urology.2021.10.053. Epub 2022 Jan 19. PMID: 35063460.

Author Response

The authors of the present study give many thanks to the reviewer for these critiques. 

The reviewer presents many excellent points and further citations which we have included. 

We re-analysed our data and collated the nephrometry scores using the RENAL nephrometry scoring system and included these results. We have further updated previous references and included the excellent citations the reviewer has recommended. 

We thank the reviewer for their feedback, changes can be seen in red for the manuscript.

Reviewer 2 Report

Line 40 – “dependent” instead of “dependant”

Line 40 – it is difficult to extrapolate the results obtained from a single surgeon to the general population of surgeons. Different surgeons, have different experiences, work in different centers, with different working methologies.

Line 47 – this should be reformulated because it is not accurate. This method doesn’t reduce inter-operator bias because there is only one operator.

Line 52 – it is a retrospective study with all the weaknesses that are know

Line 57 – Single surgeon experience is a weakness in any study

Line 98 -using “time based” metrics is not the correct way to evaluate proficiency

Line 172 – “shorter hospital stays” have no significant meaning. It might happen that having a robot makes the surgeons more enthusiastic in discharging patients sooner. Time is not a good metric.

Author Response

The authors thank the reviewer for their critiques. 

Various spelling and  grammatical changes have been corrected with many thanks.

We have reformatted the wording previously given in line 47 regarding single surgeon studies and their biases. Moreover we have taken these points into account and have altered our emphasis on the benefits of single surgeon studies which we agree have certain limitations. We have elaborated on the limitations of the single surgeon study from some of the points given in the discussion to make this clear. 

We also clarified about time based metrics not being good for proficiency as you have provided.

We thank the reviewer for their time and review of the paper and hope they feel the paper is ready for publication. We are happy to make any further changes they feel may be necessary.

Kind regards

Round 2

Reviewer 1 Report

Dear Authors, thank you for the efforts made to improve the quality of your study. 

There are still two typing errors:

- Page 3: “3tilized3zed” (line 101)

- Page 5: “5tilized” (line 149)

Author Response

Thank you, 

These changes have been made. We thank the reviewer for the responses.